# Time-Dependent Serial Changes of Antigen-Presenting Cell Subsets in the Ocular Surface Are Distinct between Corneal Sterile Inflammation and Allosensitization in a Murine Model

**DOI:** 10.3390/cells10092210

**Published:** 2021-08-26

**Authors:** Kyoung-Woo Kim, Hyun-Ju Lee, Hyeon-Ji Kim, Mee-Kum Kim

**Affiliations:** 1Laboratory of Ocular Regenerative Medicine and Immunology, Biomedical Research Institute, Seoul National University Hospital, 103 Daehak-ro, Jongno-gu, Seoul 03080, Korea; kkanssa@cau.ac.kr (K.-W.K.); dalmuly@empas.com (H.-J.L.); tuntun9@naver.com (H.-J.K.); 2Department of Ophthalmology, College of Medicine, Chung-Ang University, 102 Heukseok-ro, Dongjak-gu, Seoul 06973, Korea; 3Department of Ophthalmology, College of Medicine, Seoul National University, 103 Daehak-ro, Jongno-gu, Seoul 03080, Korea; 4Transplantation Research Institute, Seoul National University Medical Research Center, 103 Daehak-ro, Jongno-gu, Seoul 03080, Korea

**Keywords:** antigen-presenting cell, allograft, allosensitization, cornea, dendritic cell, macrophage, sterile inflammation, syngraft

## Abstract

The kinetics of antigen-presenting cells (APCs) vary depending on their resident tissues and the manner of immunization. We investigated the long-term changes in mature APC and T-cell subsets over 4 weeks in the ocular surface in murine models of corneal quiescent or potent sterile inflammation, and allosensitization using partial (PT), syngeneic (Syn), and allogeneic (Allo) corneal transplantation. In PT, CD11b^int^CD11c^hi^MHCII^hi^CD86^hi^ cells increased until 4 weeks with an increase in IFNγ^hi^ T cells. In Syn, both CD11b^int^CD11c^hi^MHCII^hi^CD86^hi^ and CD11b^hi^CD11c^hi^MHCII^hi^CD86^hi^ APC subsets increased until 4 weeks with a brief increase in CD69^hi^ T cells at 2 weeks. In Allo, CD11b^int^CD11c^hi^MHCII^hi^CD86^hi^ and CD11b^hi^CD11c^hi^MHCII^hi^CD86^hi^ APC subsets increased until 4 weeks, and an early increase in CD69^hi^ T cells was observed at 2 weeks followed by a late increase in IFNγ^hi^ T cells at 4 weeks. The frequency of the IFNγ^hi^ T cell subset was positively correlated with the frequency of the CD11b^int^CD11c^hi^MHCII^hi^CD86^hi^ subset, indicating the existence of APC–T cell interaction in the ocular surface. Together, the results indicate that allosensitization in mature APCs leads to T-cell activation in the ocular surface, whereas sterile inflammation merely induces a brief and non-specific T-cell activation in the ocular surface.

## 1. Introduction

The eye is equipped with immune privilege through immunologic ignorance and active immunosuppression [1,2]. However, under certain conditions such as dry eye disease (DED) and keratoplasty, ocular immune privilege is compromised. Although corneal transplantation is one of the most successful treatments [3], long-term graft rejection occurs in up to 40% of cases [4,5]. As a first step in antigen recognition, the cornea, retina, and choroid are endowed with antigen-presenting cells (APCs), such as dendritic cells (DCs) and macrophages [6,7].

There is robust and convincing evidence showing that corneal DCs play key roles in immune privilege, DED, neuro-immune networks, and transplantation [3,6,8,9,10,11]. The kinetics of various DCs and macrophages vary depending on the tissues involved and the method of immunization [12]. In eyes, injuries to the corneal nerve elevated the population of major histocompatibility complex class II (MHCII)^+^ or CD86^+^ mature DCs in the ocular surface, and further activated type 17 helper T (Th17) cells and CD8^+^ interferon gamma (IFNγ)^hi^ cells in draining cervical lymph nodes (dLNs) at 4 weeks after the injury [8]. In addition, desiccating injuries in the ocular surface led to an increased frequency of CD11b^−^CD11c^+^DCs, CD11c^+^CD207^+^ DCs, and macrophages within 2 weeks [9]. Detailed subset analyses of classical DCs (cDCs) and myeloid cells in non-lymphoid organs have been conducted more widely in lungs and intestines than in the eye [13,14,15].

Given that the cornea is immune-privileged, we hypothesized that early changes in APCs within the ocular surface may be different than those in other organs. A few studies have investigated the kinetics of corneal APCs in sterile inflammation [8,9,16]; however, how the change of APC subsets is distinct in sterile inflammation versus allosensitization has not been fully elucidated.

In this study, three different mouse models were used to breach corneal immune-privilege through quiescent or potent sterile inflammation and corneal allogenic transplantation. We compared the time serial changes of APC and T cell subsets in the ocular surface over a period of 4 weeks.

## 2. Materials and Methods

### 2.1. Animals and Study Design

The experiments were serially conducted at one, two, and four weeks. Balb/c female mice (*n* = 54), which were 7 weeks old, were purchased from Orient Bio Inc. (Seongnam, Gyeonggi-do, Korea). The mice cages were in a specific pathogen-free facility at Seoul National University Hospital Biomedical Research Institute, maintained at 22–24 °C with 55% ± 5% relative humidity, and given food and water ad libitum [17]. The Balb/c mice were assigned to four groups: (1) control (Ct, *n* = 3 mice), (2) partial trephination group (PT, *n* = 5 mice), (3) syngeneic (Syn, *n* = 5 mice), and (4) allogeneic (Allo, *n* = 5 mice) corneal transplantation group, with one set for each time point (*n* = 18 mice/week). All mice were sacrificed under anesthesia with zolazepam/tiletamine (Zoletil, Virbac, Carros, France). To further characterize the APC subsets, which were further purified with F4/80 from the ocular surface, an additional experiment was conducted and the mice were sacrificed at 2 weeks (*n* = 20 mice; 5 per group). We did not perform the sample size calculation in this preclinical pilot study. However, the number of each group in this study was based on a previous study that analyzed the immune cell subsets in a set of corneal transplantations in a murine model [18].

### 2.2. Mouse Model of PT or Corneal Transplantation

The cornea was partially cut using a circular punch (Kai Europe GmbH, Solingen, Germany), creating a 2.0 mm diameter lesion, as described previously [8]. For PT, quiescent sterile inflammation was induced by nerve and tissue injury.

A 2.5 mm diameter donor cornea from Balb/c or C57BL/6 mice was transplanted to a 2.0 mm diameter recipient bed in Balb/c mice using four interrupted 10–0 nylon sutures, as described previously [18]. For the syngraft, potent sterile inflammation was induced by nerve and tissue injury, ischemia–reperfusion injury, suture-induced irritation, and bleeding. For the allograft, allogenic immunization was added to the induced inflammation described above. The sutures of all grafts were not removed until sacrifice to maintain the suture-related sterile inflammation. Sutures for tarsorrhaphy were removed at 1 week.

### 2.3. Gating Strategies for APCs and T Cells Using Flow Cytometry

To evaluate the kinetics of the APC subsets at different locations, the whole cornea and circumferential bulbar conjunctiva, with an approximately 1 to 2 mm resection margin from limbus, cervical lymph nodes, and spleens, were extracted at 1, 2, and 4 weeks. Using a flow cytometer (S1000EXi Flow Cytometer, Stratedigm, San Jose, CA, USA), the proportion of APCs was determined after forward and side scatter-gating (FSC/SSC) by measuring the expressions of CD11b, CD11c, and MHCII. The serial changes in mature APCs exhibiting high expressions of MHCII, CD11b^int^CD11c^hi^MHCII^hi^CD86^hi^, and CD11b^hi^CD11c^hi^MHCII^hi^CD86^hi^ were evaluated (Figure 1), as described previously [6]. To evaluate the T cells, the proportion of CD3^+^IFNγ^hi^ or CD3^+^CD69^hi^ cells was determined [19,20]. The time-dependent kinetics of the cells mentioned above were evaluated.

To further subclassify APCs into DCs and macrophages in the ocular surface, CD45^+^ cells were first sorted according to the expression of CD11b or CD11c and were re-gated using the expression of F4/80 and MHCII based on a previous study [13].

The collected cells were immunostained with the following fluorescence-conjugated anti-mouse monoclonal antibodies: CD45 (30-F11; eBioscience, Thermo Fisher Scientific, Waltham, MA, USA), CD11b (M1/70; eBioscience), CD11c (N418; eBioscience), MHCII (M5/114.15.2; eBioscience), F4/80 (BM8; eBioscience), CD3 (145-2C11; eBioscience), CD69 (H1.2F3; eBioscience), and IFNγ (XMG1.2; eBioscience). The cells were stimulated for 18 h with 5 μg/mL anti-CD3 mAbs (145-2C11; eBioscience) and anti-CD28 mAbs (37.51; eBioscience) in the presence of brefeldin A solution (eBioscience) for intracellular staining. Data were analyzed using FlowJoTM10 software (v10.6.2; Tree Star, Inc., Ashland, OR, USA).

### 2.4. Corneal Histology for Evaluating the Distribution of MHCII^hi^-Expressing Cells

The samples from the ocular surface tissues were subjected to hematoxylin–eosin or MHCII immunostaining (NIMR-4; Abcam, Cambridge, UK) with nuclear counterstaining using 4′,6-diamidino-2-phenylindole. Fluorescent images were obtained using a microscope (DP74, Olympus, Olympus, Tokyo, Japan).

### 2.5. Regression Analysis for Evaluating the Relationship between T-Cells and APC Subsets

A simple linear regression analysis was conducted to determine the correlation between the frequency of T cells and the frequency of mature APC subsets. The frequency data of all eyes at 2 and 4 weeks from all four groups were pooled per one regression analysis (*n* = 36).

### 2.6. Statistical Analysis

Prism software v.8.1.2 (GraphPad, La Jolla, CA, USA) was used for the statistical tests. Normal distribution analysis was conducted before the selection of parametric versus non-parametric statistical methods. To compare the mean of both groups, the data were analyzed using a parametric two-tailed Student’s *t*-test, or non-parametric Mann–Whitney U test, depending on the normal data distribution. For time-dependent frequency changes, a parametric one-way analysis of variance (ANOVA), followed by Bonferroni’s post hoc analyses or the non-parametric Kruskal–Wallis test followed by Dunn’s post hoc analyses was selected depending on the normal data distribution. The compositional analyses of the APC subsets among the groups were performed using the chi-squared test. The correlation between the APC subsets and T subsets was analyzed via simple linear regression. The data are expressed as mean ± standard error measurement, and the differences were considered statistically significant at *p* < 0.05.

## 3. Results

### 3.1. Time-Dependent Serial Changes of APC and T-Cell Subsets in Response to Sterile Inflammation or Allosensitization in the Ocular Surface

First, we addressed the time-serial changes of two types of mature APC subsets, including CD11b^int^CD11c^hi^MHCII^hi^CD86^hi^ and CD11b^hi^CD11c^hi^MHCII^hi^CD86^hi^ cells, and CD3^+^CD69^hi^ and CD3^+^IFNγ^hi^ T cells in the ocular surface (Figure 1), which responded to a partial corneal trephination (i.e., PT group) as a quiescent inflammation or to a syngeneic corneal transplantation (i.e., Syn group) as a potent sterile inflammation and to allogeneic corneal transplantation (i.e., Allo group) as an allosensitization (Figure 2 and Figure 3). In the PT group, the immunofluorescence of MHCII was clearly expressed in the corneal epithelial layer at 2 weeks (Figure 2A). The frequencies of both CD11b^int^CD11c^hi^MHCII^hi^CD86^hi^ and CD11b^hi^CD11c^hi^MHCII^hi^CD86^hi^ cells increased at 4 and 2 weeks, respectively (*p* = 0.024 and *p* < 0.0001, respectively, Figure 2B). Notably, the frequency of the IFNγ^hi^ T cells increased concomitantly with the increase in the frequency of CD11b^hi^CD11c^hi^MHCII^hi^CD86^hi^ cells at 2 weeks (*p* = 0.003, Figure 2C).

The expression of MHCII^hi^ cells increased at the stroma near the graft junction in both the Syn and Allo groups (Figure 3A). In the Syn group, the CD11b^int^CD11c^hi^MHCII^hi^CD86^hi^ and CD11b^hi^CD11c^hi^MHCII^hi^CD86^hi^ subsets increased until 4 weeks (*p* < 0.05, Figure 3B), which was accompanied by an increased frequency of CD69^hi^ T cells at 2 weeks (*p* = 0.003, Figure 3C). In the Allo group, both APC subsets significantly increased up to 4 weeks (*p* < 0.05, Figure 3D), which was accompanied by increased frequencies of both T-cell subsets (*p* < 0.05, Figure 3E). At 2 weeks, an early increase in the CD69^hi^ T-cell subset was observed, followed by a late increase in the IFNγ^hi^ T cell subset at 4 weeks (Figure 3E). The proportional population of CD11b^hi^CD11c^hi^MHCII^hi^CD86^hi^ cells was much higher than that of CD11b^int^CD11c^hi^MHCII^hi^CD86^hi^ cells under both steady-state and grafted conditions (Figure 3B,D). Furthermore, time-dependent changes were greater in CD11b^hi^CD11c^hi^MHCII^hi^CD86^hi^ cells compared with CD11b^int^CD11c^hi^MHCII^hi^CD86^hi^ cells in both the Syn and Allo groups (Figure 3B,D).

### 3.2. Regression Analysis of the T-Cell Subset Proportions Related to the APC Subsets in the Ocular Surface

To evaluate whether the CD11b^int^CD11c^hi^MHCII^hi^CD86^hi^ and CD11b^hi^CD11c^hi^MHCII^hi^CD86^hi^ APC subsets interact with the CD69^hi^ or IFNγ^hi^ T cell subset, a regression analysis was conducted. In the ocular surface, the frequency of the IFNγ^hi^ T cells was positively correlated with only the frequency of the CD11b^int^CD11c^hi^MHCII^hi^CD86^hi^ cells (*r* = 0.338 and *p* = 0.044), whereas the frequency of the CD69^hi^ T cells was negatively correlated with the frequency of the CD11b^int^CD11c^hi^MHCII^hi^CD86^hi^ cells (*r* = −0.449 and *p* = 0.006) (Figure 4A,B). The frequency of the CD11b^hi^CD11c^hi^MHCII^hi^CD86^hi^ cells did not correlate with that of either of the T cell subsets (Figure 4C,D).

We further purified the CD11b^hi^CD11c^lo^MHCII^hi^ and CD11b^hi^CD11c^hi^MHCII^hi^ APCs according to the co-expression of F4/80, a well-known marker for macrophages, at 2 weeks when CD3^+^CD69^hi^ or CD3+IFNγ^hi^ cells were activated in all three groups (Figure 5A). We assumed that the CD11b^hi^CD11cloMHCII^hi^F4/80lo and CD11b^hi^CD11c^hi^MHCII^hi^F4/80^−^ subsets were mature DC-like APCs, and that the CD11b^hi^CD11c^lo^MHCII^hi^F4/80^hi^ and CD11b^hi^CD11c^hi^MHCII^hi^F4/80^+^ subsets were mature macrophage-like APCs. The subpopulation of the CD11b^hi^CD11c^hi^ cells and the subpopulation of the CD11b^hi^CD11c^lo^ cells were higher in the Syn and Allo groups regardless of the expression of MHCII and F4/80 compared with the control and the PT group (Figure 5B–G). There was no significant difference in the frequencies of all those populations between the control and PT groups, nor between the Syn and Allo groups (Figure 5B–G).

## 4. Discussion

This study suggests that corneal CD11b^+^CD11c^+^ DCs may exert their priming function differently between sterile inflammation and allosensitization. Corneal allosensitization (i.e., allogeneic corneal transplantation) resulted in the continuous increase in ocular CD11b^int^CD11c^hi^MHCII^hi^CD86^hi^ and CD11b^hi^CD11c^hi^MHCII^hi^CD86^hi^ DC subsets until 4 weeks accompanied by an activation of IFNγ^hi^ T cells. Similarly, corneal potent sterile inflammation (i.e., syngeneic corneal transplantation) resulted in the increase in CD11b^int^CD11c^hi^MHCII^hi^CD86^hi^ and CD11b^hi^CD11c^hi^MHCII^hi^CD86^hi^ DC subsets until 4 weeks; however, only CD3^+^CD69^hi^ cells increased briefly at 2 weeks. Quiescent sterile inflammation (i.e., partial trephination) increased the CD11b^hi^CD11c^hi^MHCII^hi^CD86^hi^ DC subset only at 2 weeks accompanied by a brief increase in the CD3^+^IFNγ^hi^ subset at the same time. In addition, both macrophage-like APC and DC-like APC subset populations were higher in the Syn and Allo groups compared to the PT group at 2 weeks. To the best of our knowledge, this is the first study demonstrating how the ocular APCs are activated in a time-dependent manner, following the different corneal immune responses.

Few studies have investigated the kinetics of corneal APCs during sterile inflammation [8,9,21,22]; however, most of these studies did not differentiate between APC subtypes. Under the conditions of dry eye, the frequencies of CD11b^−^CD11c^+^ and macrophages increased up to 2 weeks in the ocular surface [9]. In the PT group, the frequencies of APCs in the ocular surface increased up to 4 weeks [8]. Although both the dry eye and PT models exerted sterile inflammation, corneal DC subsets were differently activated between these groups [8,9]. With epithelial or stromal cell damage, the macrophage population likely accumulated to enhance wound healing [23]. A disease-specific increase in the CD11b^+^CD11c^+^ subset was also observed in retinal disease [7]. Taken together, APC subsets distinctly exert their function in a tissue and inflammatory-type-specific manner.

Regarding potent sterile inflammation, the frequency of ocular CD69^hi^ T cells was temporarily elevated without a change in the IFNγ^hi^ cell population. This may be due to the absence of APC–nodal T cell interaction because we discovered that there were no changes in nodal CD69^hi^ and IFNγ^hi^ T cells in the Syn group (Appendix A) despite the increase in ocular CD11b^+^CD11c^hi^MHCII^hi^CD86^hi^ DCs, whereas nodal CD69^hi^ or IFNγ^hi^ T cells increased in the Allo group (Appendix A). As resident memory T cells can be reactivated in extra-lymphoid tissues [24], ocular DCs may also prime T cells in the cornea or recruit them from the conjunctiva-associated lymphoid tissue when the danger signals trigger sterile inflammation. Unlike sterile inflammation, allo-antigen undergoes dLN imprinting nodal T activation with their migration to the ocular surface, wherein sterile inflammation predominantly induces local APC–T cell interactions, which are not from distant lymph nodes.

Because APCs represent a heterogeneous population with multiple overlapping surface markers [14], a complete differentiation among the subsets is challenging. In this study, the CD11b^+^CD11c^hi^MHCII^hi^CD86^hi^ subsets increased similarly over time in both the Syn and Allo groups. Since CD11b^+^ monocyte-derived inflammatory DCs (iDCs) can be recruited under inflammatory conditions [12], the CD11b^+^CD11c^hi^MHCII^hi^CD86^hi^ subsets may have included iDCs in the Syn group. Although iDCs are diverse and present a mixed ontogeny, sharing similar expressions in MHCII, CD11b, and CD11c to those in cCDs and iDCs, they could not be fully discriminated from cDCs with the applied limited markers [25]. Further markers, such as CC-chemokine receptor 2 (CCR2) and CD115, are required to discriminate iDCs.

In general, migratory cDCs are MHCII^hi^ with mature phenotypes, whereas resident cDCs are MHCII^+^ with immature phenotypes [14,26]. Recently, specific transcription factors were used to distinctively characterize CD11b^−^ cDC type 1 (cDC1) and CD11b^+^ cDC type 2 (cDC2) [12]. The cDC1 favors activating CD8 cytotoxic T lymphocytes (CTLs) or cross-prime Th1 cells, and the cDC2 mostly induces Th1, Th2, Th17, and T_reg_ responses [12,27]. However, it is still elusive how cDCs stimulate either Th1, Th2, or CTL responses. Different uptake techniques of antigen, lysosomal processing, antigen dose, and cytokine milieu, or where the antigen is delivered in specific zones of the dLNs affect the fate of T-cell differentiation [12]. Not only is IL-12 secreted by cDC1 required for a sustained IFNγ expression of Th1 and CTL [28], but IRF4 can also facilitate cDC2 to prime a Th1 response [29]. Stronger antigenic stimulation, such as a major MHC mismatch, favors Th1 over Th2 development [30]. As allogeneic corneal rejection is mediated primarily by Th1 or partly by either Th2 or CTL responses [31,32], CD3^+^IFNγ^hi^ cells would be Th1s or CTLs in allografted cornea. Therefore, corneal CD11b^+^ cDC2s carrying the major mismatched-MHC antigens may prime Th1 or CTL producing IFNγ.

Interestingly, we found that there is a leading population of CD11b^hi^CD11c^hi^MHCII^hi^F4/80^hi^ cells, which are suggestive of the overlapped cells between DCs and macrophages in the CD11b^hi^CD11c^hi^ subsets (Appendix A). Given that the macrophages contribute to wound healing [23,33], the frequencies of the F4/80^hi^CD11b^hi^CD11c^lo^ cells were higher in full-thickness-cut lesions (i.e., Syn and Allo groups) compared with partial-cut ones (i.e., PT group) (Appendix A).

Dry eye disease can change the inflammation profile in the cornea and conjunctiva. Although all mice were free of dry eyes with a consistent environment initially, there was a concern that surface dryness may develop during the experiment in 4 weeks due to lid deformation or an irregular ocular surface after keratoplasty. It could have affected DC activation in the ocular surface of the Syn and Allo groups. Recently, the ocular surface has been reliably assessed in a non-invasive manner using optical coherence tomography (OCT) or intravital multiphoton microscopy for dry eye, forensic science, or DC kinetics in vivo, even in mice eyes [34,35,36,37]. Indeed, the ocular surface still poses diagnostic problems after death and explanation, which, if monitored in vivo, can help to better understand the underlying physiopathology. Therefore, these noninvasive diagnostic tools will be useful in investigating the in vivo pathophysiology of dry eye disease or the kinetics of immune cells.

This descriptive study has several limitations, which include: (1) a small number of subjects, (2) the lack of an early observation time, (3) the indefiniteness of the APC characteristics as DCs, and (4) the lack of regulatory T cell evaluation. Despite these limitations, we evaluated the flow of cytometric values in each group compared to the normalized values in the normal controls; moreover, the absence of isotype control or fluorescein minus one control might detract from the accuracy of the subset population changes.

Nevertheless, our study includes several noteworthy discoveries, as follows: (1) it is the first comparative study showing that APC–T cell interactions are distinct between sterilization and allosensitization; (2) it indicates a correlation between CD11b^int^CD11c^hi^MHCII^hi^CD86^hi^ cells and ocular IFNγ^hi^ T cells. The study also provides clinically relevant information for ophthalmologists that (1) neuro-immune crosstalk exists in the cornea, thereby topical T-cell suppressant may be beneficial in mechanical corneal nerve damage and neurotrophic keratitis; and (2) the optimal time taken to maintain the anti-inflammatory treatment can be determined based on the activated T cell contraction response with regard to the sterile surgical trauma of the cornea.

In conclusion, the study outcome indicates that allosensitization activates both CD11b^int^CD11c^hi^MHCII^hi^CD86^hi^ and CD11b^hi^CD11c^hi^MHCII^hi^CD86^hi^ APC subsets in the ocular surface and primes IFNγ^hi^ T cells concomitantly, whereas sterile inflammation confines CD11b^int^CD11c^hi^MHCII^hi^CD86^hi^ or CD11b^hi^CD11c^hi^MHCII^hi^CD86^hi^ APCs to the ocular surface, predominantly inducing a non-specific T cell activation temporarily. The limited inflammation may be explained by the distinct kinetics of APCs in sterile stimuli.

## Figures and Tables

**Figure 1 cells-10-02210-f001:**
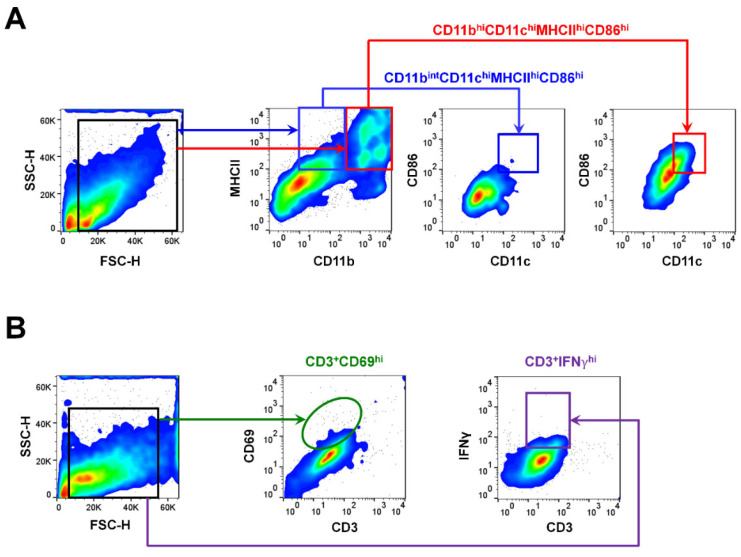
Gating strategies and the distribution pattern of the mature antigen-presenting cell (APC) and T-cell subsets. (**A**,**B**) Representative photographs of the gating strategies for the identification of APC subsets using the surface marker of CD11b, CD11c, MHCII, and CD86, and T-cell subsets using the surface marker of CD3, CD69, and interferon gamma (IFNγ) in the ocular surface.

**Figure 2 cells-10-02210-f002:**
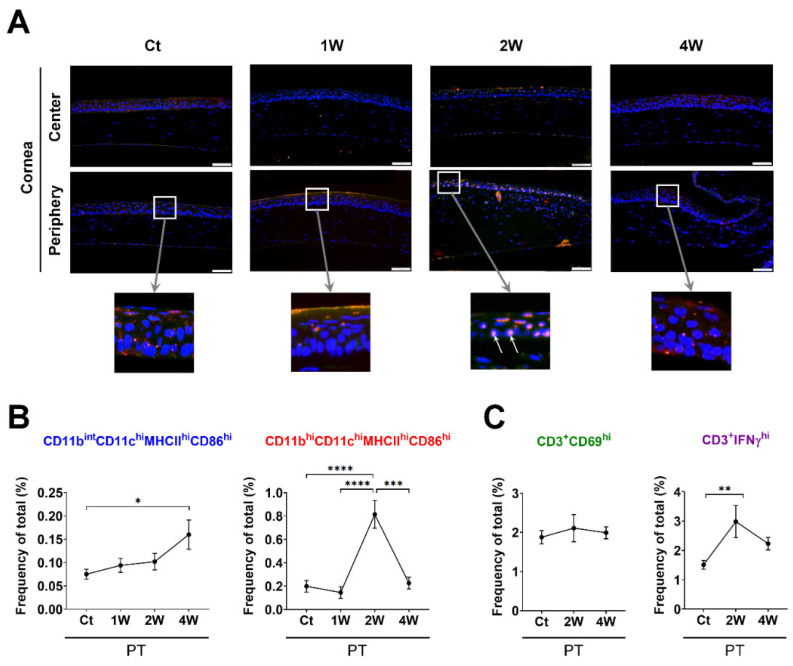
Time-dependent serial changes in APC subsets and T cells in the ocular surface in response to quiescent sterile inflammation using partial trephination until 4 weeks. (**A**) In immunofluorescent staining photographs, numerous MHCII-expressing cells were observed in the corneal epithelial layer at 2 weeks (white arrows) (scale bars: 50 μm). (**B**,**C**) Time-dependent frequency (% of total) changes in the CD11b^int^CD11c^hi^MHCII^hi^CD86^hi^ and CD11b^hi^CD11c^hi^MHCII^hi^CD86^hi^ subsets (**B**) and T-cell subsets (**C**) in the ocular surface. The proportions of CD11b^int^CD11c^hi^MHCII^hi^CD86^hi^ cells increased over time. The frequency of IFNγ^hi^ T cells increased at 2 weeks during the 4 week observation period. *n* = 5 for each time point and *n* = 12 for Ct; ANOVA followed by Bonferroni’s post hoc analyses. * *p* < 0.05, ** *p* < 0.01, *** *p* < 0.001 and **** *p* < 0.0001. Values are expressed as mean ± standard error measurement. Ct indicates the control group and PT the partial trephination group. Abbreviations: APC, antigen-presenting cell; IFNγ, interferon gamma.

**Figure 3 cells-10-02210-f003:**
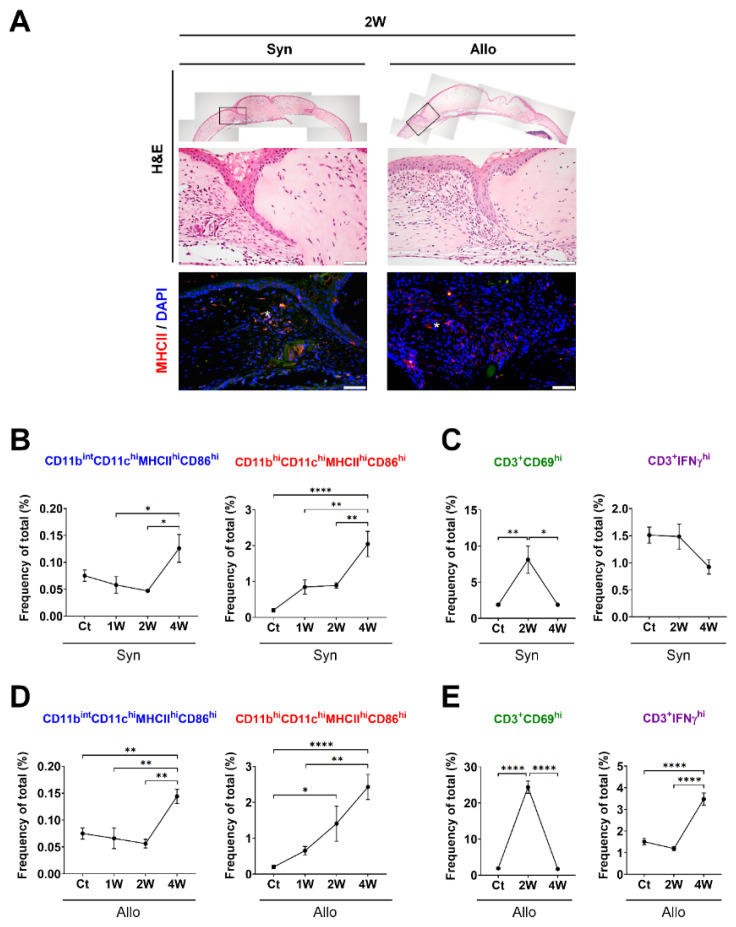
Time-dependent serial changes in APC subsets and T cells in the ocular surface in response to potent sterile inflammation or allogenic immunization up to 4 weeks. (**A**) Representative photographs of hematoxylin–eosin (H&E) and the MHCII immunofluorescent staining of the cornea. The MHCII-expressing cells (red fluorescence, w^hi^te asterisk) were found in the stromal layer near the graft junction of both the Syn and Allo groups (scale bars: 50 μm). (**B**–**E**) Time-dependent frequency changes (% of total) in the CD11b^int^CD11c^hi^MHCII^hi^CD86^hi^ and CD11b^hi^CD11c^hi^MHCII^hi^CD86^hi^ subsets and T subsets over 4 weeks in either the Syn or Allo group. The frequencies of both APC subsets increased in the Syn and Allo group over time with a priming of only CD69^hi^ T cells in the Syn group and both the CD69^hi^ and IFNγ^hi^ T cells in the Allo group. *n* = 5 for each time point and *n* = 12 for Ct; either parametric ANOVA followed by Bonferroni’s post hoc analyses or a non-parametric Kruskal–Wallis test followed by Dunn’s post hoc analyses was selected, depending on the normal data distribution. * *p* < 0.05, ** *p* < 0.01 and **** *p* < 0.0001. Ct indicates the control group; Syn, the syngeneic corneal transplantation group; and Allo, the allogeneic corneal transplantation group. Abbreviations: APC, antigen-presenting cell; IFNγ, interferon gamma.

**Figure 4 cells-10-02210-f004:**
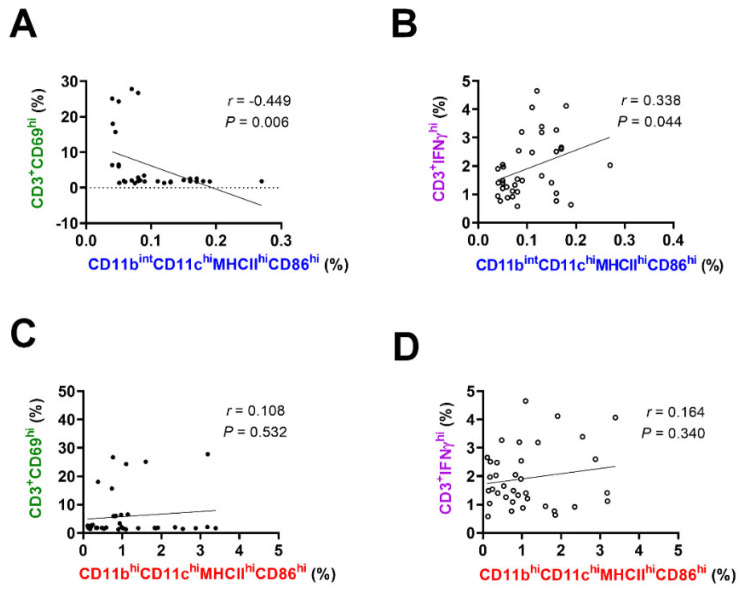
Simple linear regression analysis of the T cell proportion related to the ocular APC subsets. (**A**,**B**) The frequency (% of total) of ocular CD69^hi^ T cells was negatively correlated with the frequency of the CD11b^int^CD11c^hi^MHCII^hi^CD86^hi^ cells in the surface, whereas the frequency of IFNγ^hi^ T cells was positively correlated with the frequency of the CD11b^int^CD11c^hi^MHCII^hi^CD86^hi^ cells. (**C**,**D**) The frequency of the CD11b^hi^CD11c^hi^MHCII^hi^CD86^hi^ cells did not correlate with the frequency of T cell subsets. *n* = 36 pairs per analysis. Abbreviations: APC, antigen-presenting cell, IFNγ, interferon gamma.

**Figure 5 cells-10-02210-f005:**
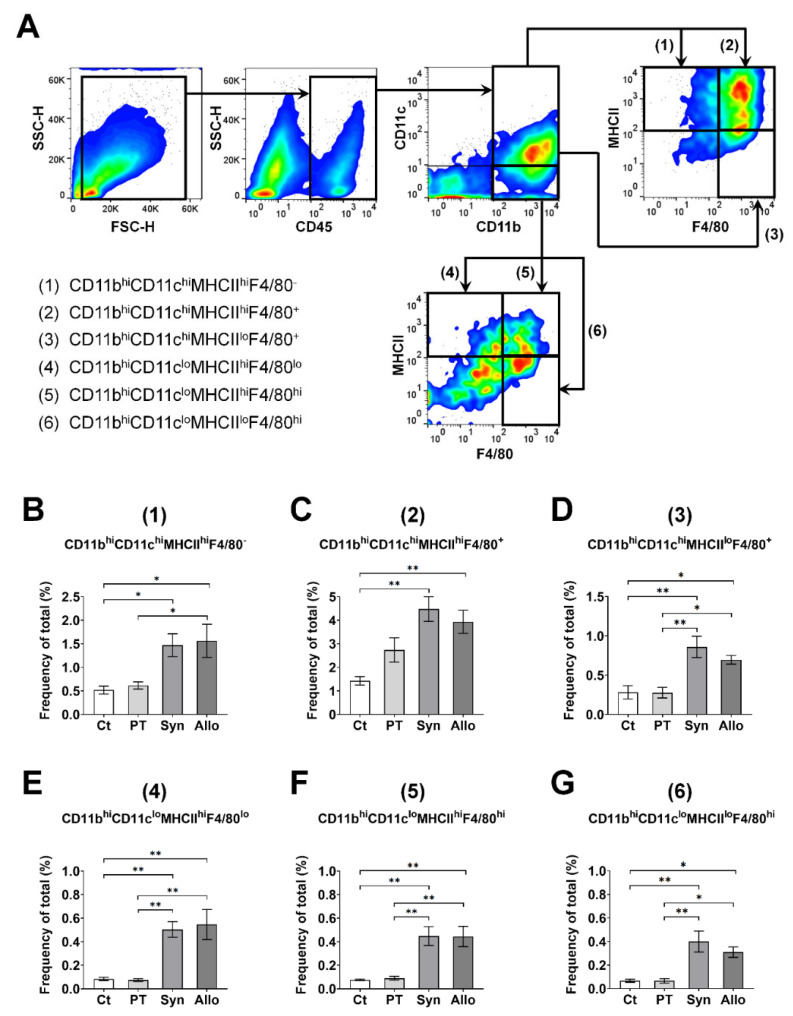
Intergroup difference in APCs with and without F4/80 expression in the ocular surface. (**A**) Representative photographs of the gating strategies to identify subpopulations of the CD11b^hi^CD11clo and CD11b^hi^CD11c^hi^ APCs according to the surface expression of MHCII and F4/80. (**B**–**G**) The frequencies (% of total) of the CD11b^hi^CD11c^hi^MHCII^hi^F4/80^-^, CD11b^hi^CD11c^hi^MHCII^hi^F4/80^+^, CD11b^hi^CD11c^hi^MHCII^lo^F4/80^+^, CD11b^hi^CD11cloMHCII^hi^F4/80^lo^, CD11b^hi^CD11c^lo^MHCII^hi^F4/80^hi^, and CD11b^hi^CD11c^lo^MHCII^lo^F4/80^hi^ subsets were significantly higher regardless of the expression of MHCII and F4/80 in the Syn and Allo groups compared with the control and PT groups. *n* = 5 for each group; ANOVA followed by Bonferroni’s post hoc analyses. * *p* < 0.05 and ** *p* < 0.01. Values are expressed as a mean ± standard error measurement. Ct indicates the control group; PT, the partial trephination group; Syn, the syngeneic corneal transplantation group; and Allo, the allogeneic corneal transplantation group. Abbreviations: APC, antigen-presenting cell.

## Data Availability

All data generated or analyzed during this study are included in this published article.

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
