# Peer review of "Time-Dependent Serial Changes of Antigen-Presenting Cell Subsets in the Ocular Surface Are Distinct between Corneal Sterile Inflammation and Allosensitization in a Murine Model"

_cells, 2021, doi:10.3390/cells10092210_

Round 1

Reviewer 1 Report

   In the present article, the authors analyzed serial changes of  antigen presenting cell (APC) subsets in the ocular surface, comparing those in the corneal sterile inflammation, syngeneic transplantation and allogeneic transplantation. Although the authors stated that there was  the difference between sterile inflammation and allogeneic transplantation, they did not mention on the similar findings observed in syngeneic transplantation and allogenic transplantation (Fig. 5).  The author should further discuss on this point.

Reviewer 2 Report

The authors present an interesting analysis of DC / macrophage subsets kinetics in corneal sterile injury compared to syngeneic and allogeneic corneal grafts. The most relevant finding is an increase of interferon+ T cells in allo grafts animals which correlates with CD11b high and intermediate CD86+ APC subsets. While the issue is of interest the work remains mainly descriptive and further analyses of T cells is lacking which would significantly improve the manuscript. All together the only relevant finding is, that in allografted animals an T cell interferon response is induced which is known already. It would significantly improve the manuscript if the authors could at least mention in the discussion potential mechanisms how these APC subsets induce the interferon response and what the potential differences are (obviously the most important difference is a major MHC mismatch) triggering the interferon response.

Minor:

Please correct the type in line 86 exhibit.

Please add the clones for the antibody in addition since catalogue number may change through time

Line 265: The LPS effects in lung are irrelevant for the article and should be taken out (citation 11). The following sentence contains common knowledge and should be taken out as well.

Overall, the discussion contains a lot of speculation and should be reduced to the discussion of the results and the specific conditions resulting in an interferon response.

Reviewer 3 Report

I read with interest the article by Kyoung Woo Kim et al. entitled “Time-dependent serial changes of antigen presenting cell subsets in the ocular surface are distinct between the corneal sterile inflammation and allosensitization” attempting to investigate the the long-term change of mature APC and T cell 14 subsets over 4 weeks in the ocular surface in murine models of corneal quiescent or potent sterile 15 inflammation and allosensitization using corneal partial trephination (PT), syngeneic (Syn), and allogeneic corneal transplantation (Allo).

However, several flaws and questions are raised in this manuscript that need to be addressed and clarified.

  1. What is the usefulness of this article? What are the real advantages of these findings for ophthalmologists? Please clarify.
  2. The title seems announce the theme of time-dependent serial changes of antigen presenting cell subsets in the ocular surface in humans. However, the aims of this study is to investigate these changes in an animal model. Probably, the authors should add in the title the following words: “in an animal model” or “in a murine model”.
  3. The article reports “Balb/c female mice (n 59 = 54) which were 7 weeks old were purchased from Orient Bio Inc. (Seongnam, Gyeonggi-60 do, Korea)”. Did the authors study the ocular surface of mice during the experiment? How environmental conditions were maintained and monitored?
  4. Please note that dry eye disease can change the inflammation profile in the cornea and conjunctiva. The authors should note that the ocular surface can be reliably assessed in a non-invasive manner by OCT as described in the following articles (also in mice!): PMID: 25369027 and 25536051. Please quote the aforementioned articles and discuss this important point for potential future studies.
  5. The introduction and the discussion should be made more robust and convincing by means of a more thorough bibliographic research. Indeed, the ocular surface still poses diagnostic problems after death and after explanation which, if monitored, can help to better understand the underlaying physio-pathology, as reported in the following article: PMID: 31321632. Please quote and discuss this issue.
  6. The methods are explained rather superficially. Authors should better explain the selection (inclusion and exclusion) of the animals. For example, what were all the criteria for defining the mice as normal (non-dry eye)? Did the authors perform the sample size calculation? Normality test? 

In sum, the article is quite interesting, but discussion should be improved as suggested. At the moment, the paper is not ready for publication. I have some concerns about some areas noted above, particularly in bibliographic research, and I would like to see the revised version again before allowing it to be published.

Round 2

Reviewer 1 Report

  The revised version of manuscript was improved according to the reviewer's comments.

Author Response

We deeply appreciate you for the supportive review.

Reviewer 2 Report

The authors significantly improved the manuscript. Unfortunately, the discussion contains a significant number of corrections mistakes. As example the sentence in line 288 does not make sense due to a error during editing. The stentence in line 314 went wrong too. In sentence 359 I would rather use the term "pathophysiology". Overall I would recommend to upload a new clear version after careful checking sind the current version is challenging.

Reviewer 3 Report

The article is now ready for publication

Author Response

(The authors gave the same response as above.)
